# Assessment of the Biological Impact of SARS-CoV-2 Genetic Variation Using an Authentic Virus Neutralisation Assay with Convalescent Plasma, Vaccinee Sera, and Standard Reagents

**DOI:** 10.3390/v15030633

**Published:** 2023-02-25

**Authors:** Naomi S. Coombes, Kevin R. Bewley, Yann Le Duff, Matthew Hurley, Lauren J. Smith, Thomas M. Weldon, Karen Osman, Steven Pullan, Neil Berry, Bassam Hallis, Sue Charlton, Yper Hall, Simon G. P. Funnell

**Affiliations:** 1Medical Interventions Group, UK Health Security Agency, Porton Down, Salisbury SP4 0JG, UK; 2Division of Infectious Disease Diagnostics, National Institute for Biological Standards, Medicines and Healthcare Products Regulatory Agency, Potters Bar EN6 3QG, UK

**Keywords:** SARS-CoV-2, SARS-CoV-2 variant, SARS-CoV-2 neutralisation assay, coronavirus, immune escape

## Abstract

In the summer of 2020, it became clear that the genetic composition of SARS-CoV-2 was changing rapidly. This was highlighted by the rapid emergence of the D614G mutation at that time. In the autumn of 2020, the project entitled “Agility” was initiated with funding from the Coalition for Epidemic Preparedness Innovations (CEPI) to assess new variants of SARS-CoV-2. The project was designed to reach out and intercept swabs containing live variant viruses in order to generate highly characterised master and working stocks, and to assess the biological consequences of the rapid genetic changes using both in vitro and in vivo approaches. Since November 2020, a total of 21 variants have been acquired and tested against either a panel of convalescent sera from early in the pandemic, and/or a panel of plasma from triple-vaccinated participants. A pattern of continuous evolution of SARS-CoV-2 has been revealed. Sequential characterisation of the most globally significant variants available to us, generated in real-time, indicated that the most recent Omicron variants appear to have evolved in a manner that avoids immunological recognition by convalescent plasma from the era of the ancestral virus when analysed in an authentic virus neutralisation assay.

## 1. Introduction

In December 2019, the first cases of a disease with unknown aetiology were reported within clusters of patients in Wuhan, China [1]. Clinical presentation of the disease was similar to that of Severe Acute Respiratory Syndrome (SARS) and Middle East Respiratory Syndrome (MERS), consisting of a severe viral pneumonia [2,3]. By the 10 January 2020, the first genome of the causative agent was published which identified it as a novel member of the *Betacoronavirus* genus closely related to SARS Coronavirus (SARS-CoV-1), subsequently officially named SARS-CoV-2 [4,5,6]. As the virus spread rapidly around the world, pandemic status was declared by the World Health Organization (WHO) on 11 March 2020 [7]. Despite multipronged interventions and mitigation approaches, SARS-CoV-2 continues to cause significant disruption to healthcare systems and economies all over the world. As of 4 December 2022, there have been approximately 641 million cases of Coronavirus disease 2019 (COVID-19) and in excess of 6 million deaths worldwide [8].

Immediately following the publication of the viral genome, several developers began work on vaccine candidates, with the first clinical trial conducted in March 2020 for the Moderna mRNA-1273 vaccine (NCT04283461). Until recently, most subsequently approved vaccines were based on the ancestral virus spike glycoprotein due to existing evidence from SARS-CoV-1 and MERS-CoV studies that spike-based vaccines induce neutralising antibodies that appear to confer significant protection to immunised recipients [9,10]. The primary mechanism of action for antibodies that neutralise coronaviruses is through binding to the spike glycoprotein. This blocks interaction with the host cell receptors, thus preventing virus entry and infection. Other proposed mechanisms of action include binding to particles and causing aggregation, or inhibiting the release of viral genomes from endosomes [11]. In August 2022, bivalent vaccines containing both ancestral and Omicron subvariant spike glycoproteins (either BA.1 or BA.4/BA.5, depending on the formulation) were also approved in both the UK and the US [12,13]. However, it remains that the majority of the population who have been vaccinated against SARS-CoV-2 received a vaccine based on the ancestral virus.

In November 2020, the first variant of concern (VOC) was described in the United Kingdom, now called Alpha [14,15]. The Technical Advisory Group on Virus Evolution, overseen by the WHO, monitors emerging variants and classifies them as variants of concern (VOCs) depending on their epidemiology, genetic composition, clinical disease presentation or effectiveness of diagnostics, vaccines, and therapeutics. To date, there have been five VOCs as classified by the WHO: Alpha (Pangolin lineage B.1.1.7), Beta (B.1.351), Delta (B.1.617.2), Gamma (P.1), and Omicron (B.1.1.529) [16]. The emergence of SARS-CoV-2 variants continues to interfere with global recovery from the pandemic and there are concerns over the emergence of resistance to convalescent or vaccine-induced immunity [17,18].

One of the most practical and recognised methods to assess escape from immunity is the viral neutralisation assay using panels of convalescent and reference plasma/sera. SARS-CoV-2 is currently classified by the UK’s Advisory Committee for Dangerous Pathogens and the US Centers for Disease Control and Prevention (CDC) as a hazard group 3 virus. As a result, studies with live SARS-CoV-2 are required to be conducted at microbiological biosafety level 3 (BSL3). Consequently, the majority of virus neutralisation data published to date have been generated in surrogate assays using pseudotyped viruses (PSVs) which, while valuable for allowing high throughput of SARS-CoV-2 data at BSL2, have several limitations. For example, PSVs typically only contain SARS-CoV-2 spike protein and are devoid of the other components of the viral pathogen’s genome, such as other structural proteins and products of the 14 open reading frames that SARS-CoV-2 possesses [19]. The spike protein is also often modified in order to increase spike expression and improve PSVs titres which is known to have varying effects on virus entry depending on cell type [20]. In addition, the morphology of PSVs often differs to that of the authentic virus. For example, the shape and size of PSVs, as well as the density and distribution of the spike protein on the viral surface, may not be representative of the authentic virus. SARS-CoV-2 is a spherical virus particle with a diameter of 70–80 nm [21]. Many of the PSVs are based on a lentivirus or vesicular stomatitis virus (VSV) background which have spherical (80–100 nm) or bullet-shaped (70 × 200 nm) virions, respectively [22,23]. Therefore, using authentic virus neutralisation tests such as the Plaque Reduction Neutralisation Test (PRNT), or the similarly based Focus Reduction Neutralisation Test (FRNT), continues to be the gold standard for measuring neutralisation antibodies [24].

The CEPI-funded Agility project was established to monitor and assess the biological risk of SARS-CoV-2 variants in order to achieve timely reporting to key stakeholders including policy makers and vaccine manufacturers. In addition to all WHO classified VOCs, we also evaluated variants with the potential to become a VOC, as identified by the WHO, CDC, and UK variant assessment committees. In this manuscript, we refer to these as variants under investigation (VUIs). Here, we describe neutralisation profiles of 20 different variants (22 isolates) with FRNT, performed in parallel at the UK Health Security Agency (UKHSA) and the Medicines and Healthcare products Regulatory Agency (MHRA) against a panel of convalescent plasma collected prior to the emergence of the Alpha variant. We also included the first WHO International Standard (IS) (NIBSC 20/136) and interim working standards (NIBSC 21/234 and NIBSC 21/338) to enable an independent comparator of antibody reactivity.

More recently, BA.4 and BA.5.2.1 sublineages of Omicron have displayed increased resistance to neutralisation by pre-Alpha convalescent plasma and the interim reference material (NIBSC 21/234). As a result, we also assessed these variants of Omicron using sera taken from volunteers who have received multiple doses of first-generation vaccines. These data suggest that BA.4 and BA.5.2.1 have drifted further away from the ancestral virus than those previously investigated. Whilst in vivo studies using the Omicron variant (BA.1) isolated in December 2021 suggest both humoral and T-cell immunity to ancestral virus cross-reacts, ongoing studies are examining the virulence of BA.4 and BA.5.2.1 using an in vivo model [25].

## 2. Materials and Methods

### 2.1. Convalescent Plasma, Vaccinee Sera, and Antibody Standards

Convalescent plasma were obtained from NHS Blood and Transplant (NHSBT). Plasma samples were obtained from 11 individuals who had recovered from a confirmed SARS-CoV-2 infection between May and June 2020. The plasma were screened using an in-house pseudovirus neutralisation assay (methods described below) prior to their admission into this study. The results of this screening assay are shown in Appendix A. Sera from participants who had received three doses of first-generation vaccine (Wuhan-based) were collected as part of the ESCAPE study as previously described [26]. This serum panel was collected at a median of 72 days post-booster, involving 10 UKHSA staff volunteers following three doses of vaccine. These participants had received an initial two-dose course of either Comirnaty (*n* = 4) or Vaxzevria (*n* = 6). All participants received an mRNA booster vaccine, and one of the participants tested positive for COVID-19 in the interim. Three standards were included in this study: NIBSC 20/136, NIBSC 21/234, and NIBSC 21/338. The WHO IS 20/136 and NIBSC 21/234 reagents are pools of several pre-Alpha convalescent plasma [27,28]. NIBSC 21/338 is a pool of plasma from 265 individuals who are both vaccinated and convalescent following an Alpha, Beta, or Delta infection [29].

### 2.2. Cell Maintenance and Media

The cells used in this study were Vero/E6 (ECACC 85020206) for assays, Vero/hSLAM (ECACC 04091501), and Vero E6-ACE2-TMPRSS2 (VAT) (NIBSC #101003) cells for virus isolation/propagation [30,31]. The cells were maintained in MEM (Gibco, Paisley, UK), 10% heat-inactivated foetal calf serum (Sigma, Gillingham, UK), 2 mM L-Glutamine (Gibco, Paisley, UK), 25 mM HEPES (Gibco, Paisley, UK), and 1× NEAA (Gibco, Paisley, UK). In addition, Vero/hSLAM cultures were supplemented with 0.4 mg/mL of geneticin (Invitrogen, Loughborough, UK) and VAT cells with 2 mg/mL geneticin and 200 µg/mL hygromycin B (Invitrogen, Loughborough, UK) to maintain the expression plasmid. For infection and cultivation media, the foetal calf serum was reduced to 0% and 4%, respectively.

### 2.3. Virus Isolation from Clinical Material

Where possible, the virus was isolated from clinical nasopharyngeal swabs taken from patients infected with a sequence-verified SARS-CoV-2 variant. Care was taken to isolate and propagate working banks of the virus on Vero/hSLAM or VAT cells to avoid selection for, and amplification of, viruses with a furin cleavage site (FCS) mutation [30,31,32]. Swab material was centrifuged at 13,000× *g* for 5 min to pellet sample debris. Vero/hSLAM or VAT cells twice-washed with DPBS (Gibco, Paisley, UK) in T12.5/T25 flasks were inoculated with 100–250 µL (dependent on available volume). Samples were adsorbed onto the cells for one hour in infection media. Cultivation media was added to the flasks and they were incubated at 37 °C without CO_2_, with daily monitoring for signs of cytopathic effect (CPE). In addition, 2× antibiotic/antimycotic (Thermo Fisher, Waltham, MA, USA) were used in all isolation media. Sterile 3 mm borosilicate beads were used to gently dissociate remaining attached cells by rocking over the monolayer, followed by clarification with centrifugation at 1000× *g* for 10 min. The aliquoted virus was stored at <−60 °C.

### 2.4. Virus Bank Propagation

Virus banks were grown on Vero/hSLAM by infecting at approximately 0.0005 MOI media for one hour. Cultivation media were added, and the cells were incubated at 37 °C without CO_2_, with daily monitoring for signs of CPE. When cells displayed clear CPE in 50–100% of the monolayer (typically on day four post-infection), the virus was harvested. Sterile 6 mm borosilicate beads were used to gently dissociate remaining attached cells by rocking over the monolayer followed by clarification with centrifugation at 1000× *g* for 10 min. The aliquoted virus was stored <−60 °C.

### 2.5. Quality Control of Virus Banks

All of the virus banks were subject to the following quality control checks and only used for neutralisation assessments if they passed these. Titrations were performed using viral plaque assay and/or virus focus forming assay as described previously [25]. Each variant was sighted into the FRNT to calculate the dilution at which approximately 130 foci per well could be counted in the non-neutralised control (NNC). Whole genome sequencing was performed to identify the variant and confirm presence of the FCS S1/S2 boundary. Briefly, extraction used the mini-RNA extraction kit (Qiagen, Manchester, UK) using the manufacturer’s instructions. The samples were purified as per the Zymo Clean and Concentrator manufacturer’s protocol (Zymo, CA, USA). Amplification was carried out using the SISPA method as previously described [33] and sequenced. Library preparations were carried out using a Nextera XT kit (Illumina, Cambridge, UK) resulting in 2 × 150 bp read lengths and sequenced using Illumina MiSeq or NextSeq via the Central Sequencing Laboratory service at UKHSA Colindale. Sequence data of the virus banks used in the neutralisation tests in this study are available in Appendix A. All Illumina data passed a Phred quality score of 30, enabling base call accuracy of 99.9%. Positive and negative controls were run on each plate. Mapping analysis was carried out using BWA-MEM v.0.7.17(r1188) (Heng Li, Hinxton, UK) [34] with default parameters against reference sequence NC_045512.2 and SAMtools v 1.11 (Heng Li, Hinxton, UK) [35] was used to sort and index BAM files before calling consensus and variant information using QuasiBAM v2.09 (an in-house C++ programme created by Richard Myers at UKHSA, Colindale, UK). Consensus sequences were called at a minimum depth of 100 although average read depth ranged from 1263 to 24,017. Variants called where present were in more than 80% of reads. Ambiguous reads were recorded where mixed bases were present in more than 20% of reads. The sterility of virus stocks was assessed after incubation for 7 days in tryptone soya and thioglycolate broths (E&O Labs) at both 20 °C and 37 °C. Virus stocks were also analysed for mycoplasma with PCR using Mycoplasma-specific PCR primers with ECACC SOP ECC73.

### 2.6. Focus Reduction Neutralisation Test (FRNT)

Neutralising antibody titres were measured with FRNT as described previously [24,25], with the following modifications. Samples were diluted 1 in 2 in duplicate over an extended range (1/20 to 1/40,960). The total incubation time was reduced from 24 h to 20 h for Beta and Gamma, or 26 h for all BA.x variants (except BA.5.2.1 which had an incubation time of 22 h). Immunostaining with anti-RBD antibodies was performed for all variants except BA.x where anti-nucleocapsid staining was used, as described previously [24,25]. Plate images were captured, and foci were recognised and counted using an ImmunoSpot S6 Ultra-V analyser with BioSpot counting module (Cellular Technologies, Bonn, Germany). Median neutralising titres (ND_50_) were determined using SoftMax Pro (SMP) v7.0.3 (Molecular Devices, CA, USA) or GraphPad Prism v9.5.0 (730) (GraphPad Software Inc., LA Jolla, CA, USA) by curve-fitting to a four-parameter logistic (4PL) nonlinear regression model. If the reference sera failed to produce a reference curve with 4PL, Probit regression analysis was performed using R v4.1.2 (R Core Team, Vienna, Austria) as previously described [24].

### 2.7. SARS-CoV-2 Pseudotyped Neutralisation Test

The SARS-CoV-2 pseudotyped lentivirus was generated with transient transfection of HEK293T/17 cells. Briefly, the cells were seeded in a 10 cm dish and transfected 24 h later with plasmids encoding the HIV-1 gag-pol genes (p8.91), a firefly luciferase reporter gene (pCSFLW), and the SARS-CoV-2 spike gene (pCAGGS SARS-CoV-2-WuhanSpike) at a ratio of 1:1.5:1 µg, using Fugene-HD (Promega, Southampton, UK) according to the manufacturer’s instructions. The supernatant was harvested at 48 h and 72 h post-transfection, and then pooled and stored at −80 °C. The neutralising activity of the convalescent plasma was assessed using HEK293T/17 cells, transfected 24 h prior to infection with plasmids encoding human ACE2 and TMPRSS2. Briefly, three-fold serial dilutions of plasma samples were mixed with 200 TCID50 of SARS-CoV-2 lentiviral pseudotyped virus for 1 h at 37 °C, at 5% CO_2_. The mixture was then added to the transfected HEK293T/17 which were seeded at 20,000 cells/well in a 96-well plate and incubated for at least 2 h at 37 °C at 5% CO_2_ before use. Following incubation of the diluted plasma and pseudotyped virus, the mixture was transferred to the target cells and incubated for 70 h at 37 °C at 5% CO_2_. Infection was measured with the detection of luciferase expression using the Promega Bright-Glo assay system and the GloMax Navigator plate reader, following the manufacturer’s instructions. Data were normalised to transduced cells and uninfected cells, and the IC90 was determined with GraphPad Prism v9.5.0 (730) (GraphPad Software Inc., LA Jolla, CA, USA), using a four-parameter logistic regression.

### 2.8. Statistics

Log_10_-transformed ND_50_ titres were analysed using a mixed-effects linear model in R [36]. The fixed effect was the virus variant (as a factor) and the random effect was the plasma sample ID. A two-way ANOVA with Tukey’s HSD post-hoc pairwise test was used to determine significance and estimates of fold change, and log estimates of fold change were back-transformed into unlogged values. The code used for analysis is available as a Appendix A.

Regression analysis with a Pearson correlation was used to compare the neutralisation titres from the PSV and FRNT assays in Minitab v19.1 (Minitab Ltd., Coventry, UK) on the log10-transformed titres.

The geometric coefficient of variation (%GCV) was calculated as appropriate when analysing log-normal geometric mean titres using the method described by Canchola et al. [37].

## 3. Results

Where possible, the virus was isolated from clinical swabs taken from individuals infected with a sequence-verified SARS-CoV-2 variant. Deep sequencing (Illumina) was performed on propagated stocks to check for cell-culture-induced mutations, whether the furin cleavage site had been maintained, and to verify the lineage. The variants used in this paper, along with their sources and other details, are listed in Table 1.

We began by assessing the neutralising ability of a panel of 11 convalescent plasma obtained between May and June 2020, before Alpha had emerged. We also included the WHO IS (NIBSC 20/136) and subsequent secondary reference material (NIBSC 21/234 or 21/338), calibrated against the IS, as the standard was depleted globally. The virus we refer to as ancestral is a pre-D614G B-clade virus (Australia/VIC01/2020) and has >99.9% sequence identity with the Wuhan-Hu-1 reference genome (GenBank: MN908947) [31]. The panel represented a range of high and low responders, as determined on the Diasorin LIAISON SARS-CoV-2 IgG, EuroImmune SARS-CoV-2 S1, and lentiviral-based PSV neutralisation assays (Appendix A). The PSV neutralisation assay used to screen these plasmas was retrospectively found to significantly correlate with the results of the ancestral virus FRNT (Pearson’s r = 0.63; *p* = 0.038).

FRNT was used to determine the plasma dilution which resulted in a 50% reduction in the number of foci (ND_50_). These data were generated in duplicate from two independent laboratories (UKHSA and MHRA). During characterisation of the variant virus stocks, differences in foci phenotype were observed. Notably, the foci for Beta were considerably larger, whereas Omicron foci were typically much smaller. This was easily accommodated with minor changes to the spot recognition protocol in the BioSpot software. Interestingly, the BA.5.2.1 foci were much larger than the foci of the other Omicron variants tested here. The incubation times were decreased or increased accordingly to permit accurate counting of approximately 130 foci in the non-neutralisation control (NNC) wells. In the majority of cases, the SARS-CoV-2 variants produced foci that were comparable to those seen for the ancestral virus. Appendix A shows where foci differed to the degree that changes to the counting parameters were required. Immunostaining for the Receptor Binding Domain (RBD) was performed as previously described, however the highly mutated RBD in Omicron (BA.x) spikes necessitated the use of an antibody against the nucleocapsid [24]. The method for this modification is presented in this manuscript.

Summaries of the neutralisation data of individual unpooled plasma are represented as the median and interquartile ranges of the titres of 11 convalescent plasma, performed in duplicate at the two independent labs, shown in in Figure 1. The IS and working reagents (WR) are represented individually as a red triangle, green box, or blue circle, respectively. Due to the exhaustion of UKHSA and MHRA stocks of reference materials, each standard was not tested against all variants. To compare the performance of the panel of individual pre-Alpha convalescent plasma, fold changes in titre against the variant relative to the ancestral virus were extracted from a linear mixed-effect regression model on the log_10_ transformed ND_50_ titres (Figure 1). The numerical breakdown of these data and the model estimates of the geometric means for the panel per variant are also presented in Appendix A.

The geometric mean (GM) of the panel against the ancestral virus was 2940. For the first VOC to be identified, Alpha, a 5.2-fold reduction in GM titre was observed with a GM of 563. The next VOC, Beta, displayed a 27.6-fold reduction with a GM titre of 106. Gamma-UKHSA was the only variant tested here to show no significant difference in neutralising titres relative to the ancestral virus with a GM of 2690. However, for Gamma-FioCruz, the GM neutralisation titre was 750, resulting in a significantly different 3.9-fold reduction relative to the ancestral virus. The difference between the two Gamma stocks was also significant with a 3.6-fold reduction (ME-ANOVA; *p* < 0.05).

The VUI labelled Alpha + E484K resulted in a GM neutralisation titre of 522 for the convalescent panel. This equates to a 5.6-fold reduction but was not found to be significantly different to the Alpha VOC from which it evolved. The largest fold reduction in neutralisation titre (39.9-fold) against the VUI was displayed when the panel was tested against the FioCruz Zeta, with a GM of 73.6. However, when tested against the Zeta sourced from BEI, the GM titre was 255, resulting in a 11.5-fold reduction. The difference in neutralisation titres against the two Zetas was statistically significant (fold change = 3.5; ME-ANOVA; *p* < 0.05).

The GM neutralisation titre of the panel against the VUI, Kappa, was 208 and resulted in a 14.1-fold reduction. Against the Delta variant, which emerged at a similar time and location as Kappa, the panel displayed a 9.2-fold reduction in neutralising titre with a GM of 318. This is not as large a reduction as was seen for Kappa, but this difference was not statistically significant. Neutralisation titres against the two Delta subvariants, AY.1 and AY.4.2 evaluated here, had a GM of 310 and 338 with 9.4-fold and 8.7-fold reductions, respectively. These were also not statistically significant from each other or from Delta. The Lambda variant, classified as a VUI, resulted in a GM neutralisation titre of 350 which represented an 8.4-fold reduction. The neutralising capacity of this panel against the Mu VUI was reduced by 32.5-fold with a GM of 90.2.

Omicron subvariants have yielded the largest fold reductions to date in neutralisation titres against the pre-Alpha convalescent panel. The GM neutralisation titres against BA.1 and BA.1.1 were 24.5 and 23.8 which correspond to 120-fold and 123-fold reductions, respectively. For BA.2 and BA.2.12.1, the GMs of the panels were 47.3 and 44.0, corresponding to reductions of 62.0-fold and 66.6-fold, respectively. The most recent Omicron subvariants tested, BA.4 and BA.5.2.1, displayed more variability in the titres, yielding GMs of 31.5 and 31.8, and reductions of 93.2-fold and 92.2-fold, respectively, relative to the ancestral virus. It should be noted that many of the titres against all sublineages of Omicron were below the lower limit of detection (LLOD) for the FRNT (ND_50_ = 20). Samples falling below the LLOD were assigned the value of 20. Furthermore, the lower limit of quantification (LLOQ) for the ancestral virus assay was determined to be 58 [24]. Although the LLOQ was not determined for subsequent variants, titres falling below this value are likely to be less precise.

As reactivity of the MHRA pre-Alpha convalescent plasma panel against the Omicron subvariants was so close to the LLOD, we also assessed the neutralisation titres of sera taken from individuals who had received three doses of a first-generation COVID-19 vaccine (Figure 2 and Appendix A). The GM titre of the vaccinee panel against the ancestral virus was 5230 (2.0-fold higher than the convalescent panel). For BA.1 and BA.1.1, the GM titres were 182 and 144, respectively, representing 29-fold and 36-fold reductions relative to ancestral virus. Against the Omicron subvariants BA.2, BA.2.12.1, and BA.2.75.3, the vaccinee panel demonstrated 26-fold, 31-fold, and 31-fold reductions with GM neutralisation titres of 199, 166, and 168, respectively. The neutralisation titres were reduced further against BA.4 and BA.5.2.1, with GMs of 104 and 65, representing fold reductions of 50 and 80, respectively. The difference between BA.4 and BA.5.2.1 was significant (fold change = 1.6; ME-ANOVA; *p* < 0.05) and both resulted in significantly lower neutralisation titres against other Omicron subvariants tested here.

Another group of variants which generated interest was the recombinants of SARS-CoV-2. We obtained clinical samples for the recombinants XE and XF and propagated the virus for these. The recombinant XE is composed of BA.1/BA.2 with a breakpoint in nsp6 (nucleotide position 11,537) resulting in a BA.2 spike sequence. The recombinant XF is a Delta/BA.1 combination with a breakpoint in nsp3 (nucleotide position 5386), resulting in a BA.1 spike sequence. We tested both the convalescent (Figure 1) and/or vaccinee panels (Figure 2) against these variants. When tested against the vaccinee panel, XE resulted in a GM neutralisation titre of 162, thus a 32-fold reduction relative to the ancestral virus. For the recombinant XF against the convalescent panel, we observed a 118-fold reduction relative to the ancestral virus with a GM of 25. The vaccinee panel resulted in a GM of 151 against XF, corresponding to a 35-fold reduction relative to the ancestral virus. In each panel, the difference between the recombinants and their parent spikes (BA.2 for XE, or BA.1 for XF) was not significant.

Two variants tested against the vaccinee panel were not included against the convalescent panel (BA.2.75.3 and the XE recombinant). After testing 19 variants (21 isolates) against the convalescent panel, the number of plasmas giving responses at or below the LLOD led to testing against the convalescent panel being discontinued in favour of testing all new variants against the vaccinee panel only.

One of the participants in the vaccinee panel had a breakthrough infection in January 2022. To assess the impact of including this participant, we performed the analysis including and excluding the sample from this participant (Appendix A). This process did not result in a significant impact on the calculated fold changes. All fold changes when this sample was excluded (*n* = 9) were within the confidence intervals calculated for the full panel (*n* = 10).

The neutralisation titres of the WHO IS 20/136 and/or the working reagents for the anti-SARS-CoV-2 immunoglobulin, NIBSC 21/234 or 21/338, were also analysed in every assay at both the UKHSA and MHRA. These are displayed in Figure 1 and Figure 2 as red triangles, green squares, or blue circles, respectively. The WHO IS 20/136 and NIBSC 21/234 reagents are pools of several pre-Alpha convalescent plasma [27,28]. NIBSC 21/338 is a pool of plasma from 265 individuals who were both vaccinated and convalescent following an Alpha, Beta, or Delta infection [29]. All variants were typically more susceptible to neutralisation with these reagents. However, it is notable that the neutralising capacity of 21/234 was greatly reduced for the Omicron subvariants BA.4 and BA.5.2.1, with fold reductions relative to the ancestral virus of 420 and 450, respectively. In comparison, the reduction for BA.1 was 60-fold. The IS was developed as a way to increase the harmonisation of results between laboratories. Figure 3A shows the neutralising titres as ND_50_ for the WHO IS 20/136 and the panel of 11 plasma against the ancestral virus assessed in triplicate at each laboratory. We examined the effect of normalisation to the WHO IS 20/136 on the titres between laboratories by normalising the titres to IU/mL (Figure 3B). The geometric coefficient of variation (GCV) was calculated to assess the variability of the assay, with the ND_50_ data yielding a GCV of 40.4% and the normalised IU/mL data producing a GCV of 22.5%. This represents an improvement in inter-lab variability of 17.9% (*p* < 0.0001). However, we also wanted to consider the effect of normalisation among the variants and subsequently calculated the normalised titres relative to the WHO IS for each variant tested against the IS (Figure 4). Here, the normalisation distorted the observed fold changes, and differences among the variants were no longer significant. This highlights the fact that the WHO IS should not be used to compare results among the variants; instead, it can be used to harmonise data from different assays and laboratories against each variant, separately.

## 4. Discussion

There is concern that SARS-CoV-2 variants will lead to immune escape, resulting in new waves of infection, hence prolonging the pandemic [38,39]. Here we attempt to address the question of SARS-CoV-2 immune escape by tracking aspects of humoral immunity to both VOCs and VUIs as they arose. By using a consistent method for isolating the virus and a panel of convalescent plasma from early in the pandemic, it was possible to determine the relative reduction in neutralising antibody titres against 19 variants (21 isolates) using an authentic virus neutralisation assay [24,32]. An initial screening of these plasma in a PSV assay (with the ancestral spike) was also performed. This showed a positive correlation to the ancestral virus FRNT and is comparable to correlations between a different PSV assay and this FRNT published previously [24]. While all variants tested demonstrated a reduction in neutralisation titres compared to the ancestral virus with FRNT, the largest reduction was seen in Omicron and its sublineages. This reduction seemed to occur in a mostly chronological manner, whereby the appearance of successive variants was characterised by concomitantly lower neutralisation activity, although this relationship was not absolute.

Of the 23 isolates (21 variants) tested, 17 were isolated in-house on Vero/hSLAM cells to reduce the risk of working stocks containing unwanted mutations or deletions [32]. For two variants (Gamma and Zeta), we tested the virus obtained from two different sources which were isolated on different cell lines. Sequence analysis revealed that the two stocks had the same spike sequence but some Single Nucleotide Polymorphisms (SNPs) at other positions within the genome with uncertain implications. A significant difference was seen between the neutralisation titres of the antibody panel against the virus from these two working stocks of the same variant. This highlights the importance of consistency in isolation techniques when aiming to compare neutralisation titres against different SARS-CoV-2 variants. The reason for this disparity is unclear; however, the S protein of SARS-CoV-2 is known to be highly glycosylated with approximately 40% of the trimer covered with glycans. This glycan shield is known to protect the virus from humoral and cellular immunity [40]. It is also possible that differences in glycosylation machinery between cell lines could result in different glycan structures on the S protein, thus resulting in different neutralisation titres depending on the cell line used for virus propagation [41,42,43].

It has also been postulated that cells infected with SARS-CoV-2 secrete free spike protein that is not associated with a virus particle [44]. It is therefore conceivable that free spike could bind to neutralising antibodies and reduce the pool available to inhibit viable virus particles. SARS-CoV-2 is also known to generate defective interfering (DI) particles. These are defective viral genomes (DVGs) that have retained the packaging signal and are able to be packed into virus particles expressing spike when a cell is co-infected with a viable helper virus [45]. Similar to the concept of free spike protein binding functional neutralisation antibodies, DIs could bind antibodies and prevent their action on viable virus. Whilst care was taken to grow our virus stocks in a consistent manner and with a low MOI to minimise the risk of DIs, it is possible that the amount of free spike or DIs present in each stock could vary. This may help to explain some of the discrepancies we observed here compared to others in the literature, and further highlights the importance of generating virus stocks in a consistent manner. These observations suggest that when comparing results between laboratories, the method of virus isolation should also be taken into consideration as well as the type of neutralisation assay. Notably, the studies reported here were conducted at two independent laboratories using the same virus propagation techniques, generating highly comparable datasets.

Caution should also be applied when comparing results with other groups due to differences in assay methodologies, and in the different use of an “ancestral” virus for derivation of titre ratio differences (fold reductions). Despite the need for caution, our results largely agree with previously published data for Alpha, Beta, Delta, and Omicron for both the authentic virus and PSV assays [46,47,48,49,50]. Large reductions in neutralisation titres have been reported by others for Beta and Omicron, although typically not as large as the reductions presented here [47,50,51].

A persistent issue with titrating panels of sera in authentic virus neutralisation assays against variants is the lower limit of detection of the assay. In our own assay, the titres of antibodies in the convalescent panel (from the pre-Alpha phase of the pandemic) were so reduced against the currently circulating virus (Omicron and its derivatives), that many of the individual plasma samples fell below the LLOD of the assay. It was for this reason that we obtained a panel of vaccinee sera for assessing the Omicron subvariants. For these vaccinee sera, the neutralising titres were above the LLOD, and most were also above the LLOQ, giving us greater confidence in our calculated titres and fold changes. We demonstrated that Omicron is less resistant to immunity induced by vaccination with smaller fold reductions in neutralisation compared to the convalescent panel. Our results were similar to other published data, with all Omicron subvariants resulting in significantly lower neutralisation titres when compared to the ancestral virus, but with little variation among the subvariants, with the exception of BA.4 and BA.5.2.1 [52,53].

One of the participants in the vaccinee panel had a breakthrough infection. No data are available to identify the variant with which the individual was infected. However, the date of the infection was in January 2022. The dominant variant in circulation in the United Kingdom at this time was Omicron BA.1 [54]. It is therefore reasonable to assume that the breakthrough infection was with this variant. Removing data for this individual had no significant impact on the fold changes. It is arguably more representative to analyse panels of sera including samples from individuals with a breakthrough infection. The majority of people in the world will at some stage experience a breakthrough infection now that nonpharmaceutical interventions in most countries have been removed [55]. The use of further vaccine boosts in the general population may also be discontinued by most healthcare systems [56].

The two Gamma isolates tested here show a statistically significant difference in neutralising titres, despite possessing the same spike sequence and only one other SNP in the rest of the genome. We observed a fold reduction in neutralising titres of the panel against one isolate but not in the other. There are examples within the literature which report similar outcomes [46,47,48,50,51]. These data further demonstrate that the variation in neutralisation results can be influenced by aspects of the SARS-CoV-2 virus other than those encoded by the spike sequence.

There are fewer examples of neutralisation results against the VUIs, however our results are largely in agreement with existing data. Although actual titres are not necessarily comparable, largely due to inevitable inter-lab variability and differences in assays, the patterns seen in reductions in titres are similar. For example, groups that have investigated neutralisation against Mu typically show that titres are similar or slightly lower than Beta, as we do here [50,57]. Similarly, sera tested against AY.4.2 and AY.1 have been shown to have similar neutralisation titres to their parent strain, Delta [58,59]. Neutralisation data against B1.1.7 + E484K are limited, but one group showed a comparable fold reduction (3.8-fold) as we see here (5.0-fold) but with vaccinee, instead of convalescent, sera [60]. For Kappa and Lambda variants, our results are in disagreement with the literature as others show a much smaller fold reduction in neutralisation titres: approximately a 1.8-fold reduction compared to our 14.1-fold reduction. However, the published results to date have utilised PSV assays for these variants which have their limitations and may not always correlate with the results from authentic virus assays [61,62].

Of the variants tested here, Zeta showed the largest discrepancy in results and further demonstrates the value of performing neutralisation tests with an authenticated virus. Here, two stocks isolated in different ways led to statistically different results despite having identical spike sequences. Zeta has only three mutations in the spike protein yet resulted in a 40-fold or 10-fold reduction relative to the ancestral virus in our hands, depending on virus source. This is in contrast to some PSV data for neutralisation titres against Zeta, which indicated that there was less potential for immune evasion with smaller fold reductions than observed in this study [63]. Another group demonstrated smaller fold reductions for Zeta than we report here, using an authentic virus assay with sera from in vivo studies using hamsters. Here, sera from hamsters infected with the ancestral SARS-CoV-2 (D614G) demonstrated only a 2-fold drop in neutralising antibodies against Zeta, relative to D614G. However, data for other VOCs tested here are also in disagreement with the literature; for example, neutralisation titres against Delta appeared to show an increase relative to those against D614G, when most evidence within the literature suggests approximately 4-fold to 8-fold reductions [47,48,57,64]. Interestingly, another group performing authentic virus neutralisation assays with early pandemic human sera showed a similar fold reduction in neutralising antibodies as we report here. However, when tested against vaccinee sera, this large fold change was no longer apparent [51]. The discrepancy between our Zeta authentic virus neutralisation results and those generated using a PSV assay further highlight the importance of verifying data using virus which contains the whole complement of viral proteins and not just spike.

As part of this study, we also compared the effect of normalising neutralisation titres to an international standard, WHO 20/136, a pool of pre-Alpha convalescent serum. We demonstrated that appropriate use of an IS permits the meaningful comparison of results between laboratories using the same virus variant. Although the IS was typically more cross-reactive to the variants, it also demonstrated a reduction in titres against the variants comparable to individual sera. It would therefore not be correct to use the standard for comparison of titres between different variants, as this practice would distort the fold changes in many instances. If normalisation between variants is required, a standard which is equally potent against the variants needs to be generated, perhaps by including a pool of convalescent sera from multiple variants, or a broadly reactive pool of well-defined monoclonal antibodies.

In summary, we report neutralising titres against the largest selection of authentic virus variants (isolated in a consistent manner) to date using an authentic SARS-CoV-2 FRNT. Caution should be taken when comparing results with other laboratories and the value of using an authentic virus in neutralisation assays has been emphasised. We also demonstrate that the use of an IS and reporting neutralisation titre in IU/mL increases comparison of results between groups for an individual variant but should not be used to harmonise data between different variants. With the emergence of Omicron and its sublineages, pre-VOC panels are unlikely to provide insightful data going forward and should be replaced with a more epidemiologically relevant panel. All activity on this project including existing and any new data will be freely available at the Agility project website [65].

## Figures and Tables

**Figure 1 viruses-15-00633-f001:**
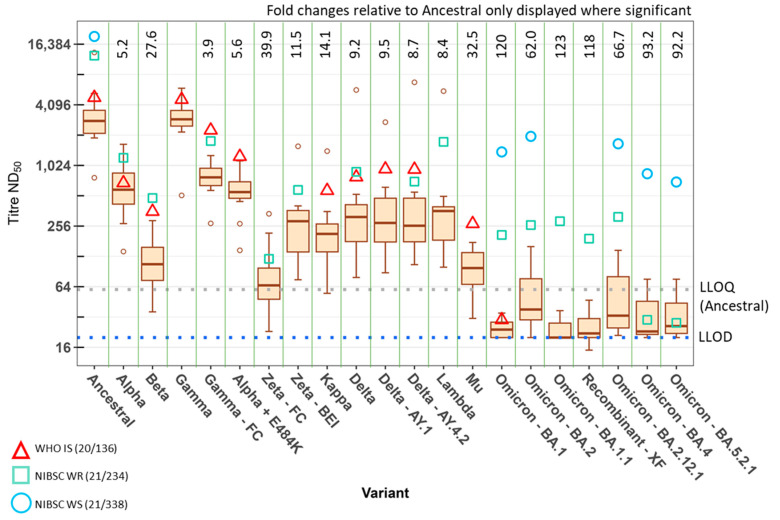
Neutralising antibody titres of a panel of pre-Alpha convalescent plasma determined with FRNT at two independent laboratories against authentic SARS-CoV-2 variants. A panel of 11 pre-Alpha convalescent plasma were assessed at UKHSA and MHRA with FRNT over a dilution range of 1/20 to 1/40,960 with 20 authentic SARS-CoV-2 variants. Data are presented as median and interquartile ranges of titre ND_50_ and are ordered by date of variant emergence. Fold changes relative to ancestral virus were calculated with regression analysis using a linear mixed-effect model and are presented where the difference is significant (two-way ANOVA with Tukey’s post hoc test). The lower limit of detection (LLOD) of 1/20 is indicated by the blue dotted line. The lower limit of quantification (LLOQ) of 1/58, as determined for the ancestral assay, is indicated by the grey dotted line. Neutralisation titres of the WHO IS 20/136, NIBSC WR 21/234, and 21/338 are presented as a red triangle, green box, or blue circle, respectively.

**Figure 2 viruses-15-00633-f002:**
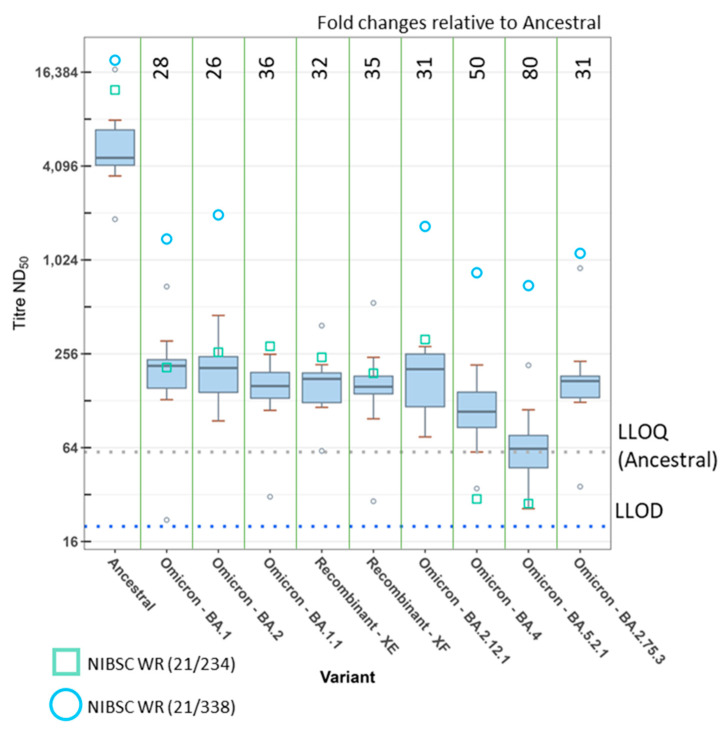
Neutralising antibody titres of a panel of triple-vaccinated human participant sera determined with FRNT against authentic SARS-CoV-2 variants. A panel of 10 sera from triple-vaccinated human volunteers was assessed at UKHSA with FRNT over a dilution range of 1/20 to 1/40,960 with 10 authentic SARS-CoV-2 variants. Data are presented as median and interquartile ranges of titre ND_50_. Fold changes relative to the ancestral virus were calculated with regression analysis using a linear mixed-effect model and are presented where the difference is significant (two-way ANOVA with Tukey’s post hoc test). The lower limit of detection (LLOD) of 1/20 is indicated by the blue dotted line. The lower limit of quantification (LLOQ) of 1/58, as determined for the ancestral assay, is indicated by the grey dotted line. Neutralisation titres of the NIBSC WR 21/234 and 21/338 are presented as a green box and a blue circle, respectively.

**Figure 3 viruses-15-00633-f003:**
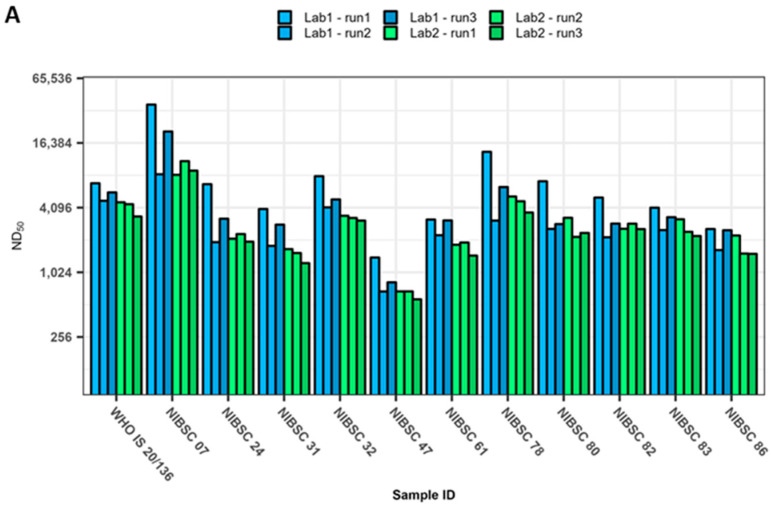
Comparison of the effect of using the WHO International Standard (IS) 20/136 for normalisation between labs. A panel of 11 pre-Alpha convalescent plasma and the WHO IS 20/136 were assessed for their neutralising capacity against authentic ancestral SARS-CoV-2 using a FRNT assay. Neutralising titres are presented as (**A**) ND_50_ or (**B**) IU/mL with normalisation to the WHO IS 20/136.

**Figure 4 viruses-15-00633-f004:**
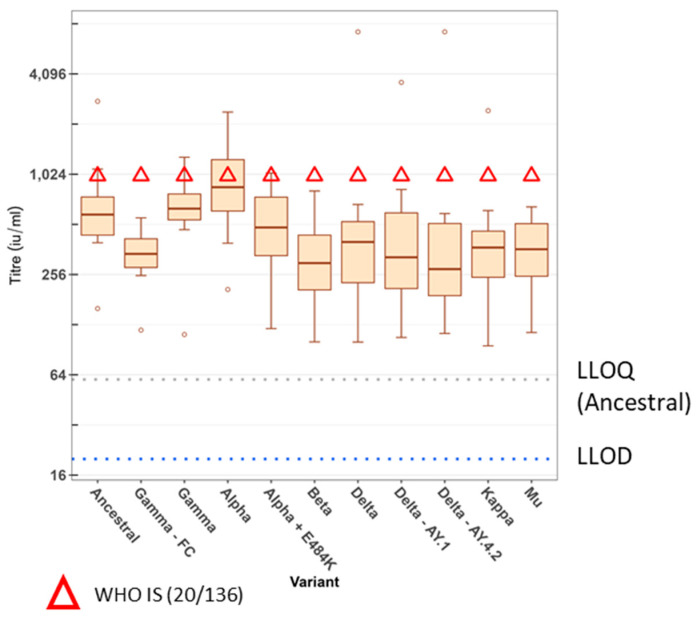
The effect of normalising antibody neutralising titres to the WHO International Standard (IS) 20/136 when comparing among SARS-CoV-2 variants. A panel of 11 pre-Alpha convalescent plasma and the WHO IS 20/136 were assessed for their neutralising capacity against authentic ancestral SARS-CoV-2 and a selection of variants using a FRNT assay. Neutralising titres are presented as IU/mL through normalisation to the WHO IS 20/136.

**Table 1 viruses-15-00633-t001:** SARS-CoV-2 variants used in this study.

WHO Designation	Pangolin Lineage	GISAID Clade/Lineage	Nextstrain Clade	Date Assigned	GISAID ID of Isolation Swab (Where Known) and/or Source
Ancestral virus	B	O	19A	Jan 2020	EPI_ISL_406844 [31]
Alpha	B.1.1.7	GRY	20I (V1)	Dec 2020	EPI_ISL_683466
Alpha + E484K	B.1.1.7	GRY	20I (V1)	Feb 2021	Gavin Screaton (University of Oxford)
Beta	B.1.351	GH/501Y.V2	20H (V2)	Dec 2020	EPI_ISL_770441
Gamma *	P.1	GR/5017.V3	20J (V3)	Jan 2021	EPI_ISL_2080492
Gamma–FioCruz *	P.1	GR/5017.V3	20J (V3)	Jan 2021	FioCruz, Brazil
Delta	B.1.617.2	G/452R.V3	21A	May 2021	EPI_ISL_2742236
Delta	AY.1	GK	21A	May 2021	EPI_ISL_2742878
Delta	AY.4.2	GK	21A	May 2021	EPI_ISL_4306633
Lambda	C.37	GR/452Q.V1	21G	June 2021	BEI (NR-55654)
Kappa	B.1.617.1	G/452R.V3	21B	Apr 2021	EPI_ISL_2742167
Mu	B.1.621	GH	21H	Aug 2021	Not available
Zeta–FioCruz *	P.2	GR/484K.V2	20B/S.484K	Mar 2021	FioCruz, Brazil
Zeta–BEI *	P.2	GR/484K.V2	20B/S.484K	Mar 2021	BEI (NR-55439)
Recombinant–XE	XE	GRA	Recombinant	Mar 2022	EPI_ISL_11586931
Recombinant–XF	XF	GRA	Recombinant	Mar 2022	EPI_ISL_10458256
Omicron	BA.1	GRA	21K	Dec 2021	EPI_ISL_7400555
Omicron	BA.1.1	GRA	21K	Jan 2022	EPI_ISL_8165999
Omicron	BA.2	GRA	21L	Dec 2021	Not available
Omicron	BA.2.12.1	GRA	22C	Apr 2022	Gavin Screaton (University of Oxford)
Omicron	BA.2.75.3	GRA	22D	Jun 2022	EPI_ISL_13882158
Omicron	BA.4	GRA	22A	Apr 2022	EPI_ISL_13157810
Omicron	BA.5.2.1	GRA	22B	Apr 2022	EPI_ISL_12810908

* For two variants, Gamma and Zeta, assays were conducted with stocks derived from viruses of different sources.

## Data Availability

All virus bank sequence data provided in the Appendix A. R scripts for data analysis provided in Appendix A. Raw data available upon request.

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
