# Peer review of "Assessment of the Biological Impact of SARS-CoV-2 Genetic Variation Using an Authentic Virus Neutralisation Assay with Convalescent Plasma, Vaccinee Sera, and Standard Reagents"

_viruses, 2023, doi:10.3390/v15030633_

Round 1

Reviewer 1 Report

The authors examined convalescent sera, sera from vaccinated persons and reference sera (consisting of a serum pool) for neutralising SARS-CoV-2 antibodies. For this purpose, 23 different SARS-CoV-2 isolates were used as antigens for the neutralisation test. Most of these viruses, corresponding to an original pre-VOC strain as well as representatives of all known VOCs and different VOC sublines, were isolated from respiratory material on a Vero cell line for the present study and characterised by complete sequencing. Each serum was tested in duplicate at different dilution levels. The neutralising properties of the sera were demonstrated by inhibition of a virus-induced cytopathic effect. Specificity and investigator independence of results are ensured by immunostaining and automated spot detection. The tests were carried out in two laboratories. As expected, convalescent sera from the beginning of the pandemic showed significantly reduced neutralising properties against seven Omicron strains. This effect was also observed in the sera of triple vaccinated individuals, where as many as 9 different Omicron isolates were used as antigens. This is not surprising either, since the vaccinations were carried out with a vaccine that had not yet been adapted to Omicron.

Testing of reference sera, including the WHO standard, all consisting of convalescent serum pools, has shown that the results are comparable between the two laboratories. In this respect, such pools are suitable as a reference to harmonise data obtained with different assays and in different laboratories. However, the same antigen must have been used.

This manuscript describes a very carefully conducted study. I appreciate that the data were obtained in neutralisation tests using 23 different SARS-CoV-2 strains as antigens. A large proportion of these strains were isolated and characterised in the present work under largely identical conditions. To date, the virus neutralisation test is the gold standard for determining humoral virus immunity. Data collected with surrogate neutralisation tests or with pseudovirus-based neutralisation tests must be measured against it. I also find it very commendable that the tests were carried out in two different laboratories.

From my point of view, there are only a few points of criticism or suggestions for improvement.

For example, the summary is teeming with abbreviations that are not listed. Furthermore, it does not present the most important results and conclusions, but comes across as very descriptive and explanatory.

Another point concerns the inconsistency in the number of antigens listed, sometimes 21 variants are mentioned (summary), sometimes 20 variants (page 2, line 90). In Table 1, however, there are 23 different isolates.

In the supplement, the authors also show data collected with a pseudovirus neutralisation test based on the spike protein of a pre-VOC strain. Unfortunately, I lack a classification of the results. From my point of view, it would be nice if the titres could be directly related to the titres obtained with a pre-VOC strain in the neutralisation test.

The description of the variability of plaque sizes in different viral strains is interesting, but the examples are somewhat singled out; it would be important to show this for all the strains used, so that one can better classify this finding.

Tables 2 and 3 contain the information already presented in Figures 1 and 2. In this respect, both tables are dispensable or could be moved to the supplement.

I also do not know why two Omicron sublines/recombinants were included in Figure 2 with XE and BA.2.75.3, which are missing in Figure 1.

Another point of criticism concerns the sera from triple-vaccinated individuals. Was a breakthrough infection excluded in these individuals?

Reviewer 2 Report

In this study neutralization of the major virus variants that have emerged during the pandemic by pre-VOC convalescent sera and sera from vaccinated individuals was tested. As described previously by several groups, neutralizing capacity of these sera was reduced to several VOC and especially to the different omicron variants. Although not really new, this study was well controlled as independent laboratories performed the tests in parallel and internal standards were used for normalization. The study indeed nicely proves, what would be expected, that using the WHO internal standard decreases interlab variations. 

However not all artifacts may have been eliminated. The fact that the two zeta as well as the two gamma isolates, where the isolate pairs differed only in a few SNP outside the S protein, differed significantly in their susceptibility to neutralization, may be indicative of such artifacts. The speculation of the authors that viral proteins other than the surface proteins may be involved in virus neutralization is highly speculative. Potential artifacts should be discussed. Was the virus stock directly used in the neutralization assay again sequenced? If not and possible, this should be done. Did the viruses primarily grow to similar titers or did the dilutions of the virus stocks for the neutralization assay differ between the viruse isolates. Were the virus stocks produced with a standard moi and harvested at the same time post infection, i.e. can it be excluded that the ratio of shed S protein or inactive virions to infectious particles differed between the virus stocks. These points should at least be discussed in detail before postulating that SNP outside the envelope glycoproteins determine the susceptibility of a virus to neutralization. Moreover, I am not aware of data in the literature that may support such an hypothesis. If there is, the authors should cite these studies. 

Round 2

Reviewer 1 Report

The authors have revised the manuscript. I consider the authors' scientific contribution to be extremely important.

From my point of view, two minor questions remain open.

The supplementary data, which were also collected comparatively with a pseudovirus neutralisation test, are not briefly discussed.

It also remains open with which virus variant the breakthrough infection took place and to what extent the resulting hybrid immunity influences the results. Since only 10 triple-vaccinated persons were examined, this should be checked again and, if necessary, factored out.
